# Simplifying Schistosome Surveillance: Using Molecular Cercariometry to Detect and Quantify Cercariae in Water

**DOI:** 10.3390/pathogens11050565

**Published:** 2022-05-10

**Authors:** Brooke A. McPhail, Kelsey Froelich, Ronald L. Reimink, Patrick C. Hanington

**Affiliations:** 1School of Public Health, University of Alberta, 357 South Academic Building, 116 St. and 85th Ave., Edmonton, AB T6G 2R3, Canada; bmcphail@ualberta.ca (B.A.M.); kfroelic@ualberta.ca (K.F.); 2Freshwater Solutions LLC, 137 W 15th St., Holland, MI 49423, USA; reiminkron@gmail.com; 3Saint Joseph High School, 2521 Stadium Dr., Saint Joseph, MI 49085, USA; 4Office of Campus Ministries, 110 E. 12th St. Hope College, Holland, MI 49423, USA

**Keywords:** cercariometry, digenean trematode, avian schistosome, cercariae, qPCR

## Abstract

Avian schistosomes are considered a public health nuisance due to their ability to cause swimmer’s itch when accidentally encountering humans rather than their intended avian hosts. Researchers have been monitoring their presence and abundance through snail collections and cercariometry. Cercariometry methods have evolved over the last several decades to detect individual schistosome species from a single water sample, simplifying the monitoring of these parasites. This methodological evolution coincides with the development of the field of environmental DNA (eDNA) where genetic material is extracted from environmental samples, rather than individual organisms. While there are some limitations with using molecular cercariometry, notably the cost and its inability to differentiate between life cycle stages, it substantially reduces the labor required to study trematode populations. It also can be used in complement with snail collections to understand the composition of avian schistosomes in an environment.

## 1. Introduction

The group of digenean trematodes known as avian schistosomes belongs to the family Schistosomatidae. Members of this group, similar to other species of schistosomes, infect a snail as the first intermediate host in their lifecycles. Avian schistosomes can cause an irritating ailment known as swimmer’s itch or cercarial dermatitis when cercariae accidentally penetrate humans, rather than their intended avian hosts [1,2,3]. This group of trematodes is related to human schistosomes, which cause the devastating disease schistosomiasis, affecting 229 million people across Asia, Africa, and South America [4,5]. Control efforts for both human [6,7,8,9,10,11,12] and avian [1,13,14,15,16,17] schistosomes have been undertaken with varying degrees of success. These include the application of molluscicides [6,15], chemotherapy with Praziquantel (humans [8,12] and birds [14,16]), water treatment [10], and definitive host relocation [17].

Researchers have been monitoring schistosomes for the better part of the last century. Avian schistosomes are considered an inconvenience and a public health nuisance due to their ability to cause swimmer’s itch [1,18,19]. However, research revealed that they may be a larger public health issue than once thought, as one species, *Trichobilharzia regenti*, can enter the central nervous system of mice [20,21]. Schistosomulae were found within the spinal cords, brain stems, and cerebellums of mice [20,21], along with the medulla oblongata [20], and infected mice were observed to have decreased motor functions [20,21]. It has been considered that not all avian schistosome cercariae are able to penetrate human skin, or if they can, they then die within the skin (causing the dermatitis) [1]. Now, with the information provided by researchers on the ability of *T. regenti* to invade the central nervous system of mice [20,21], further studies are critical to determine the fate of other species of avian schistosomes after penetrating human skin.

Due to the reliance of most digenean trematodes on a snail host, snail collections and patent infection assessments have been the main method of characterizing the trematode community in a particular environment. Snails are typically collected at the field site of interest, brought back to a laboratory, and placed in water to see what cercariae emerge [17,22,23,24,25,26]. The presence of cercariae is confirmed using a stereomicroscope [24,25]. Cercariae can generally be identified to the family level based on morphology, but further identification requires the use of molecular tools. Snails can also be dissected to determine if any trematode sporocysts are present, as cercarial shedding only detects patent infections and not those that are in a pre-patent stage, meaning that infection prevalence results could be underestimated if we rely on quantifying patent infections alone [1,7,23,27,28].

Researchers adopted the method of cercariometry, counting the number of cercariae present in a water sample, as a means of monitoring the schistosomes present in an area. Much of this research focused on enumerating cercariae of human schistosomes in the water due to the danger they pose to human health. This was originally performed by filtering and dying the cercariae and counting them using a microscope [22,28,29,30,31,32,33]. Now, our genetic understanding of avian schistosomes has advanced to a point that it allows us to track single schistosome species using the molecular tool quantitative polymerase chain reaction (qPCR).

## 2. Cercariometry Methods Past and Present

Prior to cercariometry, one method employed to detect schistosome cercariae in water was by using rodent (e.g., mouse [34,35] and hamster [33]) immersion. The rodents were floated in the water in cages for several hours in areas with suspected schistosome-infected snails [33,34,35]. They were then taken back to a laboratory where researchers would wait between two to six months [33,34,36] to determine if any exposed rodent had developed an infection by confirming the presence of adult worms. This method took far too long to obtain information on schistosome presence/absence at the site of interest. As such, there was a necessity for a faster, more informative method that could identify areas positive for schistosome cercariae [19].

Cercariometry as a technique for monitoring schistosomes has experienced many methodological advancements. These include filtration (used in both manual and molecular cercariometry) [8,17,19,22,24,25,28,29,30,31,32,33,34,36,37], positive phototropism [38], overlay technique [39], centrifugation [40,41,42], and finally qPCR [15,17,19,24,25,37,43,44,45,46]. In the early years of using cercariometry, most of the focus was on human schistosomes, and many studies highlighted the need for fast, sensitive, and specific tests for use on a larger scale [7,19,33,44,47,48,49,50]. Most recently, we have been tracking avian schistosomes in recreational waters to monitor, predict, and understand swimmer’s itch. And, we can now quantify the number of cercariae present within an environmental water sample and even determine species composition.

The filtration technique was first implemented to perform schistosome cercariometry, and it is still being utilized with modern cercariometry. Initially, water would be filtered using a plankton net [22,31,32,33,36] or filter paper [17,24,25,29,30,34,37] with a pore size small enough to collect cercariae. In some instances, formalin would be added to the sample to kill and preserve the cercariae [22,28,32,42]. In the case of manual cercariometry, different dyes would be added to stain the cercariae and make them easier to count [22,28,29,30,31,32,33,34,36,51]. These included Lugol’s iodine [30,31,32,34,51], ninhydrin [29], and light green in 2% acetic acid [22,33,36].

Positive phototropism exploited the sensitivity/attraction to light demonstrated by schistosome cercariae [38,41,52]. The cercariae were held in the dark for a minimum of 15 min, after which they were exposed abruptly to a bright light in a vertical orientation. The cercariae swam upwards and rested just below the surface where they could be recovered [38]. Klock [38] created a positive phototropism device intended for use in the field, and it exhibited a 98% cercarial recovery rate in laboratory tests. This apparatus was not widely adopted because it was bulky and inconvenient for field use.

The overlay technique developed by Sandt [39] used the positive phototropism phenomenon explored by Klock [38], while also attempting to combat the effect of water turbidity on the recovery of cercariae—a common problem with environmental samples [30,39,45,48,49]. This method involved an overlay chamber with turbid water on the bottom and clear water above. The chamber was exposed to light and the clear overlay water was then drawn off and filtered. This method was tested under lab conditions and exhibited a low rate of cercarial recovery in turbid water, leading the author to surmise that this method would not be suitable in field conditions [39].

Next, a centrifugation method was developed where water was passed through a funnel into a device that pulled cercariae to the bottom of a tube using centrifugal force [40,41,42]. In some instances, a formalin-picric acid solution was added to the tubes to preserve and stain the cercariae so that they could be counted using a microscope [42]. In other studies [40,41], dye was added to the samples after they were washed into a petri dish post-collection. The centrifugation method could be used even where water is turbid [42], as the cercariae could be dyed before they were counted, making them easier to identify.

Polymerase Chain Reaction (PCR) is often used in this context as a detection tool rather than a quantification tool [2,47,50,53,54,55]. Methods using PCR to identify schistosomes include the detection of schistosomes from water [2,54], snail [47,53], and plankton [47] samples, as well as distinguishing between cercariae that are difficult to tell apart morphologically [50]. More recently, the focus has been on qPCR [15,17,19,24,25,37,45], a sensitive and specific molecular tool that uses primers to amplify a target region of DNA and a fluorescent probe to measure amplification of the target region [19,25]. This technique has become the gold-standard for use in recreational water monitoring for enteric bacteria [56] to detect indicator organisms that inform us about the risk associated with using a specific recreation area [57]. Furthermore, microbial source tracking allows us to identify contamination sources from recreational water [58]. A parallel can be drawn from this application of qPCR and its use in the context of cercariometry where we can test water samples using an 18S pan-avian assay to determine if avian schistosomes are present [19,24]. We can also quantify the number of avian schistosome cercariae present in the sample using this targeted assay. This assay can then be used in complement with species-specific assays that allow for the identification of species from a single water sample using cytochrome C oxidase subunit 1 (CO1) [17,24,25]. This simplifies the process considerably and these samples can be preserved and retroactively analyzed (e.g., for the presence of a newly discovered or invasive species).

## 3. *Trichobilharzia* as a Model System for Human Schistosomes

Species of *Trichobilharzia* constitute the majority of the avian schistosomes [59,60] and are an excellent model system for human schistosomes. Avian schistosomes have been under investigation in Michigan since 1928, where their ability to cause swimmer’s itch was confirmed [61,62]. This was the beginning of a contentious relationship between riparians, schistosomes, and the avian hosts that deposit them in Michigan lakes (notably *T. stagnicolae* and the common merganser) [17]. The swimmer’s itch problem in the state has been well-established [13,14,15,17,24,25,61,62,63,64,65,66,67,68,69], with research encompassing different control methods [14,15,17], new species discoveries [68], swimmer’s itch incidences at different lakes [66,67,69], development of molecular methods to track avian schistosomes [17,24,25], and monitoring avian schistosome prevalence in snails [65]. As such, a unique opportunity to study these parasites is presented.

A pan-avian schistosome qPCR assay was developed based on the 18S gene [19]. This tool was implemented for swimmer’s itch research in northern Michigan as a means of quantifying cercariae from water samples [15,17,24,25]. Additionally, five species-specific assays were developed by Rudko and colleagues [17,24,25] to track the most common species of avian schistosome present in northern Michigan lakes: *Trichobilharzia stagnicolae*, *T. physellae*, Avian schistosome sp. C, *Anserobilharzia brantae*, and *T. szidati*.

## 4. Current and Future Applications

Environmental DNA (eDNA) is a broad term that encompasses intracellular or extracellular DNA taken from the environment (e.g., soil or water samples) rather than from individual organisms [44,70,71,72]. Although still evolving, this field has already expanded the versatility within environmental sampling programs. This method is also cheaper and requires less labor than traditional methods [25,37,44,45,71]. Since no collections of organisms are required, fewer samples are needed [72]. Furthermore, experts are not required for field collections and community partners can be trained to properly collect an environmental sample [44]. eDNA sampling and analyses are also particularly useful if the organism of interest is found in low quantities or is challenging to detect with traditional sampling methods [70].

Collecting snails to determine their infection status has been the main method of monitoring schistosome infections for decades. This is a useful approach when the goal is to characterize the entire trematode population in an area, but it becomes cumbersome when focusing on one or a few species of trematode, particularly if those species are rare and there is an uneven snail host distribution [25,70]. Avian schistosomes also frequently exhibit a low infection prevalence in snails [17,24,26,46,47], and the results of snail assessments often capture only one snapshot in time. Snail collections also have to be timed strategically because trematode infections take approximately one to two months to become patent [1]. Furthermore, it is nearly impossible to differentiate between species of avian schistosome cercariae using morphology alone [19,48,55,59,73]. Therefore, if the snail infection method is used to determine which avian schistosomes are present in an area, cercariae need to be identified using molecular barcoding. This involves several time-consuming steps (DNA extraction, PCR amplification, gel electrophoresis, PCR clean up, sequencing, and finally aligning sequences and searching for matches using bioinformatics software), and it is not necessary to perform qPCR on an environmental sample for most of them.

In a recent study completed in northern Michigan assessing the success of common merganser (*Mergus merganser*) relocation on the abundance of *Trichobilharzia stagnicolae* at 4 lakes, molecular cercariometry was used over two years, incorporating 30 sites and 20 collection days each year to assess the abundance of *T. stagnicolae* and two other species of avian schistosomes (*T. physellae* and Avian schistosome sp. C) [17]. Conservatively, if this study had used the snail collection method to assess program success rather than molecular cercariometry, nearly a quarter of a million snails would have been collected and removed to achieve the scope captured with molecular cercariometry, assuming 200 snails/site/sample day. This would not have been feasible, even considering the support from community volunteers that participated in sample collection [17], which demonstrates the value of using this tool.

Understanding snail infections is useful for obtaining a broad ecological understanding of a snail-trematode community, while molecular cercariometry is better suited for targeted applications. With the advancements that have been made in molecular biology, there are now more options available, and we no longer need to rely solely on snail surveys. Cercariometry allows researchers to obtain a faster and more accurate representation of the trematode community in an area when compared to snail sampling alone and can identify species of trematodes that are present at a low density [7,51]. We recognize that neither method is mutually exclusive, but should be used in complement and tailored to the research question.

With recent advances in environmental DNA techniques, we will be able to assess the biodiversity of an area using trematodes as indicator species, rather than undertaking costly and time-consuming traditional biodiversity assessments. By using molecular cercariometry rather than the snail collection method, we are not removing thousands of snails from an ecosystem. Additionally, we only need to extract DNA from environmental samples, rather than from hundreds of snail specimens, which would allow for the collection of more samples that encompass a wider area [45].

Although researchers have been using molecular cercariometry to monitor schistosome cercariae, this method could also be applied to track any trematode that has a cercarial stage. qPCR is sensitive enough to differentiate between members of the same genus within a water sample [17,25]; therefore, the detection of individual species of different genera from the same water sample is possible. This approach could be especially useful during restoration projects to track a select few “indicator” trematodes or trematodes that inform about other host species present in the environment. This could be particularly true for trematode communities consisting predominately of host-specialists.

## 5. Cercariometry Challenges

Some have proposed that not all cercariae present in the water will have infective potential [33,35,43,48]. Two studies focusing on human schistosomes [33,35] compared the difference between cercariometry and rodent immersion. The authors found that when cercariae were present, the rodents became infected [33,35], although the worm load in the rodents did not always exhibit a linear relationship with cercarial abundance in the water [35]. When cercariae were not observed, the rodents did not become infected. Even if some cercariae are not infective (e.g., because of their age [35]), the parasite is present in the water, which can also constitute useful information [43].

Others have suggested that cercariometry methods (manual or molecular) could be difficult to implement on a large scale because they require experts to carry them out in order to be successful [48,74,75]. However, recent research [17,75] has found that a community-based monitoring approach can be effective. Such programs involve community members that are familiar with the area and are invested in the health of the study ecosystems. Data suggests that there can be high reproducibility achieved between the results from community partners and an expert laboratory when performing qPCR to monitor for the presence of different hazards present in water, including avian schistosome trematodes, cyanobacteria, and HF183 Bacteroides [75].

There have also been some concerns about the cost of using qPCR to monitor trematodes as the reagents and equipment required are expensive [55]. While the materials required to perform snail collections are inexpensive and can be reused repeatedly, qPCR requires many consumables that can be costly [43,44]. Unfortunately, no supplies required to carry out qPCR can be reused due to the nature of the technique. The costs associated with undertaking these methods has decreased and will continue to decrease [24]. This is especially pertinent if we consider the lower effort required for a qPCR study versus a snail collection study, where the collection of hundreds of snails is often required on every collection day.

qPCR is also unable to differentiate between cercariae that are alive or dead or inform us as to which life cycle stage is present in the water. This has been a major critique of this method, especially where human schistosomes are concerned. Researchers are unable to determine whether the lifecycle stages that are present have been deposited by a human host (miracidia) or whether they are infective to humans (cercariae) [44]. However, cercariae are produced through asexual amplification in the snail and are released in large quantities [47,51,74], so this may be less of a concern as it is highly likely that there are more cercariae present in the water than miracidia. Furthermore, cercariae are more likely to be influenced by wind and water movement [17,69], while miracidia are more likely to be mobile within the water [76,77]. Additionally, the cercariae of some avian schistosomes are positively phototactic (they rise to the surface of the water column), and in some cases, they are able to be separated from the miracidial stage for avian schistosome species in which the miracidia are positively geotactic (move to the bottom of the water column) [78]. As such, water sampling techniques can and have been developed to target and enrich cercariae [17]. However, there are examples of avian schistosomes that exhibit a negatively geotactic miracidial stage, notably *T. regenti* [79] and *T. szidati* [1,13], which can complicate the isolation of a pure cercariae sample.

Another challenge that has been considered is that eDNA can only be used to determine the presence or absence of human schistosomes in an area [44]. However, we are able to quantify avian schistosome cercariae using the 18S gene [24], and research on human schistosomes has shown a relationship between the number of cercariae present and the concentration of eDNA [44]. Therefore, it is likely possible to obtain parasite density using eDNA. Furthermore, although we are not currently able to differentiate between cercariae and miracidia present in the water, as discussed above, the presence of miracidia is also important information. This means that the parasite is present and could be able to complete the life cycle assuming that the correct host is present [44], and sampling can be performed in a manner to favor the collection of cercariae, rather than miracidia [17].

Lastly, there is the issue surrounding the length of time DNA can be detected using eDNA methods. This could be especially pertinent for cercarial stage as they can generally live in the water for 24 h [1,13,44]. In a laboratory study, DNA from human schistosome cercariae that had decayed were detectable for 8 days after they were shed from the snails [44]. In a natural environment, this could vary due to the conditions (e.g., water temperature and movement), and studies are required to assess how the detection of schistosomes using eDNA methods differs in still and running waters [44].

## 6. Conclusions

Cercariometry methods have advanced greatly from when they were first implemented. We can now differentiate between species from a simple water sample, requiring less labor than traditional methods, with results available relatively quickly. Snail collection and analysis protocols continue to provide value, but we can employ these methods together in many cases to understand community dynamics on a much broader scale. Finally, it is more vitally important now to track the presence and abundance of avian schistosomes, since there could be public health consequences associated with human-avian schistosome encounters beyond what we have traditionally believed. Some species that initially appear to be less of a concern for cercarial dermatitis, such as a newly discovered putative avian schistosome [68], may pose greater health risks to humans than previously thought. Future research efforts will focus on this evolving conundrum.

## Data Availability

Not applicable.

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
