# Peer review of "Simplifying Schistosome Surveillance: Using Molecular Cercariometry to Detect and Quantify Cercariae in Water"

_pathogens, 2022, doi:10.3390/pathogens11050565_

Round 1

Reviewer 1 Report

A very competent review and discussion of cercariometry methods based on the comprehensive relevant literature. Researchers dealing with the transmission of schistosome cercariae will appreciate the objective evaluation of the usefulness and limitations of the eDNA methods.

Author Response

Reviewer comment 1: A very competent review and discussion of cercariometry methods based on the comprehensive relevant literature. Researchers dealing with the transmission of schistosome cercariae will appreciate the objective evaluation of the usefulness and limitations of the eDNA methods.

Author reply: Many thanks for your comments.

Reviewer 2 Report

The manuscript presented for review provides an important description of the application of the new swimmer's itch risk quantification method. The only remarks that arise concern too briefly treating the problem in the manuscript which is "review".
In line with the comments contained in the text, I suggest developing descriptions that in their current form more closely resemble short communication

Author Response

Reviewer comment 1: The manuscript presented for review provides an important description of the application of the new swimmer's itch risk quantification method. The only remarks that arise concern too briefly treating the problem in the manuscript which is "review".

In line with the comments contained in the text, I suggest developing descriptions that in their current form more closely resemble short communication.

Author reply 1: Thank you for your comments and suggestions. Areas of text were expanded upon to add more detail that we think addresses the issue raised by reviewer 2.

Reviewer comment 2: Lines 40-41: In my opinion, more details about the quoted work should be placed here. Such information strongly justifies the necessity to undertake research on avian schistosomes.

Author reply: More information on this study was added to this section to highlight the importance and potential implications of these findings.

Reviewer comment 3: Line 52: The quantitative underestimation resulting from the presence of sporocysts and not formed cercariae in snails will not be eliminated by the suggested method. In my opinion, the fragment about sporocysts can be taken out of the text.

Author reply: Edited this sentence for clarity.

Reviewer comment 4: Line 110: Due to the fact that the presented manuscript is of the review type, I suggest presenting the content of the cited articles in more detail. Inserting 7-8 quoted items in one row as an illustration of a statement is, in my opinion, too much of a shorthand. I suggest providing a few details that the authors read in the cited works, instead of a simple list.

This note covers several places in the manuscript.

Author reply: Points in the manuscript where several publications were cited were elaborated upon to provide more detail.

Reviewer 3 Report

This is an interesting and useful review paper on the environmental monitoring for schistosomes written from the perspective of researchers who work in the field of avian schistosomiasis.  As a consequence, it provides a bridge between the largely non-intersecting fields of schistosomiasis in which the primary definitive hosts are humans versus avian species.

The authors provide a concise review of the field of cercariometry which has never received much attention in the literature on human schistosomiasis despite having considerable and increasing relevance in recent years as endemic levels have decreased due to both intervention programs and general rural development.  The paper focuses on recent advances in monitoring techniques based on environmental DNA and nicely outlines both its advantages and remaining challenges. 

If there is a weakness in the paper it is a tendency to underemphasize the remaining challenges to the widespread use of eDNA monitoring in human schistosomiasis control programs in the developing world. That issue is more thoroughly described in the Discussion section of their reference 42 by Sengupta et al

Author Response

Reviewer comment 1: This is an interesting and useful review paper on the environmental monitoring for schistosomes written from the perspective of researchers who work in the field of avian schistosomiasis.  As a consequence, it provides a bridge between the largely non-intersecting fields of schistosomiasis in which the primary definitive hosts are humans versus avian species.

The authors provide a concise review of the field of cercariometry which has never received much attention in the literature on human schistosomiasis despite having considerable and increasing relevance in recent years as endemic levels have decreased due to both intervention programs and general rural development.  The paper focuses on recent advances in monitoring techniques based on environmental DNA and nicely outlines both its advantages and remaining challenges.

If there is a weakness in the paper it is a tendency to underemphasize the remaining challenges to the widespread use of eDNA monitoring in human schistosomiasis control programs in the developing world. That issue is more thoroughly described in the Discussion section of their reference 42 by Sengupta et al.

Author reply: Thank you for your comments. More information on challenges associated with eDNA for the monitoring of human schistosomes was added. This point was not elaborated on extensively because  we wanted the the focus to remain on avian schistosomes to align best with the theme of this special edition. We have expanded on this topic slightly.